# A general framework to support cost-efficient fecal egg count methods and study design choices for large-scale STH deworming programs–monitoring of therapeutic drug efficacy as a case study

Luc E. Coffeng[1]*, Johnny Vlaminck[2], Piet Cools[3], Matthew Denwood[4], Marco Albonico[5], Shaali M. Ame[6], Mio Ayana[3,7], Daniel Dana[3,7], Giuseppe Cringoli[8], Sake J. de Vlas[1], Alan Fenwick[9], Michael French[9,10], Adama Kazienga[2], Jennifer Keiser[11,12], Stefanie Knopp[11,12], Gemechu Leta[13], Leonardo F. Matoso[14,15], Maria P. Maurelli[8], Antonio Montresor[16], Greg Mirams[17], Zeleke Mekonnen[7], Rodrigo Corrêa-Oliveira[14], Simone A. Pinto[14], Laura Rinaldi[8], Somphou Sayasone[18], Peter Steinmann[11,12], Eurion Thomas[19], Jozef Vercruysse[2], Bruno Levecke[2]*

1 Department of Public Health, Erasmus MC, University Medical Center Rotterdam, Rotterdam, The Netherlands, 2 Department of Translational Physiology, Infectiology and Public Health, Ghent University, Merelbeke, Belgium, 3 Department of Diagnostic Sciences, Faculty of Medicine and Health Sciences, Ghent University, Ghent, Belgium, 4 Department of Veterinary and Animal Sciences, University of Copenhagen, Denmark, 5 National Health System, Turin, Italy, 6 Laboratory Division, Public Health Laboratory-Ivo de Carneri, Chake Chake, United Republic of Tanzania, 7 Jimma University Institute of Health, Jimma University, Jimma, Ethiopia, 8 Department of Veterinary Medicine and Animal Production, University of Naples Federico II, Naples, Italy, 9 Schistosomiasis Control Initiative, Department of Infectious Disease Epidemiology, St Mary's Campus, Imperial College London, London, United Kingdom, 10 RTI International, Washington District of Columbia, United States of America, 11 Swiss Tropical and Public Health Institute, Allschwil, Switzerland, 12 University of Basel, Basel, Switzerland, 13 Ethiopian Public Health Institute, Addis Ababa, Ethiopia, 14 Laboratory of Molecular and Cellular Immunology, Research Center René Rachou—FIOCRUZ, Belo Horizonte, Brazil, 15 Nursing school, Federal University of Minas Gerais, Belo Horizonte, Brazil, 16 Department of Control of Neglected Tropical Diseases, World Health Organization, Geneva, Switzerland, 17 Techion Group Ltd, Dunedin, New Zealand, 18 Lao Tropical and Public Health Institute, Ministry of Health, Vientiane, Lao People's Democratic Republic, 19 Techion Group Ltd, Aberystwyth, United Kingdom

☯ These authors contributed equally to this work.
* l.coffeng@erasmusmc.nl (LEC); bruno.levecke@ugent.be (BL)

## Abstract

### Background

Soil-transmitted helminth (STH) control programs currently lack evidence-based recommendations for cost-efficient survey designs for monitoring and evaluation. Here, we present a framework to provide evidence-based recommendations, using a case study of therapeutic drug efficacy monitoring based on the examination of helminth eggs in stool.

### Methods

We performed an in-depth analysis of the operational costs to process one stool sample for three diagnostic methods (Kato-Katz, Mini-FLOTAC and FECPAK[G2]). Next, we performed

**Data Availability Statement:** All relevant data are within the manuscript, its Supporting Information files, and a preceding paper (https://doi.org/10.1371/journal.pntd.0007471).

**Funding:** LEC acknowledges funding from the Dutch Research Council (NWO, grant 016.Veni.178.023). JV was financially supported through an International Coordination Action of the Flemish Research Foundation. This study and PC were financially supported by a grant from the Bill and Melinda Gates foundation (OPP1120972, PI is BL, www.starworms.org). The collection of the Ethiopian National Mapping data (which was overseen by GL) that were used to define endemicity scenarios was financially supported by the Schistosomiasis Control Initiative and the Partnership for Child Development, both based at Imperial College London, and by the Children's Investment Fund Foundation, The End Neglected Diseases Fund, and UKAID-DFID. The funders had no role in study design, data collection and analysis, decision to publish, or preparation of the manuscript.

**Competing interests:** I have read the journal's policy and the authors of this manuscript have the following competing interests: the FECPAKG2 technology was produced and distributed by Techion Group Ltd, of which ET is an employee and GM is managing director. Both hold stocks in Techion Group Ltd. The Mini-FLOTAC device is a commercial product distributed by GC, LR and MPM through the University of Napoli Federico II. All other authors declared that they have no competing interests.

simulations to determine the probability of detecting a truly reduced therapeutic efficacy for different scenarios of STH species (*Ascaris lumbricoides*, *Trichuris trichiura* and hookworms), pre-treatment infection levels, survey design (screen and select (*SS*); screen, select and retest (*SSR*) and no selection (*NS*)) and number of subjects enrolled (100–5,000). Finally, we integrated the outcome of the cost assessment into the simulation study to estimate the total survey costs and determined the most cost-efficient survey design.

## Principal findings

Kato-Katz allowed for both the highest sample throughput and the lowest cost per test, while FECPAK$^{G2}$ required both the most laboratory time and was the most expensive. Counting of eggs accounted for 23% (FECPAK$^{G2}$) or $\geq$80% (Kato-Katz and Mini-FLOTAC) of the total time-to-result. *NS* survey designs in combination with Kato-Katz were the most cost-efficient to assess therapeutic drug efficacy in all scenarios of STH species and endemicity.

## Conclusions/significance

We confirm that Kato-Katz is the fecal egg counting method of choice for monitoring therapeutic drug efficacy, but that the survey design currently recommended by WHO (*SS*) should be updated. Our generic framework, which captures laboratory time and material costs, can be used to further support cost-efficient choices for other important surveys informing STH control programs. In addition, it can be used to explore the value of alternative diagnostic techniques, like automated egg counting, which may further reduce operational costs.

## Trial Registration

ClinicalTrials.gov NCT03465488

---

### Author summary

Large-scale deworming programs are implemented worldwide to reduce morbidity caused by intestinal worms. As these programs operate in resource-poor-settings, it is important that their operational costs are minimized without jeopardizing the quality of decision-making. We present a framework for evidence-based recommendations for cost-efficient decision-making in deworming programs, using monitoring of therapeutic drug efficacy via stool examination as a case study. To this end, we first assessed the time and the cost of processing stool samples in a laboratory according to different diagnostic methods. Then for each diagnostic method, survey design, and a range of settings (predominant worm species and pre-treatment infection levels), we calculated the probability of correctly detecting a truly reduced therapeutic drug efficacy and the associated operational costs. Generally, the estimated operational costs varied across diagnostic method, survey design, worm species and disease endemicity. Based on our findings, we conclude that the use of the current diagnostic standard, the Kato-Katz method, is justified to assess drug efficacy, but that a change in survey design is warranted. Our methodology, which leverages detailed data on laboratory time and material costs, can also be used to provide evidence-based recommendations for other types of decisions in large-scale deworming programs.

## Introduction

Soil-transmitted helminths (STHs; *Ascaris lumbricoides*, *Trichuris trichiura* and the hookworm species *Necator americanus* and *Ancylostoma duodenale*) infect approximately 800 million individuals across the world and are responsible for the loss of more than three million disability-adjusted life years annually [1,2]. To control morbidity associated with these infections, the World Health Organization (WHO) strives to reduce the prevalence of moderate-to-heavy intensity (MHI) infections to less than 2% [3]. To reach this goal, anthelmintic drugs are periodically distributed to at-risk populations through large-scale deworming programs–so-called preventive chemotherapy [4]. In these programs, periodic follow-up surveys are conducted to determine whether the therapeutic efficacy of the administrated drugs is still satisfactory [5], and whether stopping or scaling down drug administration is justified [6]. However, as these programs often operate in resource-poor settings, it is important to minimize operational costs without jeopardizing the correctness of the program decisions (e.g., avoiding prematurely scaling down of preventive chemotherapy or continuing the administration of anthelmintic drugs with a reduced therapeutic drug efficacy). An important proportion of STH survey costs is related the processing of stool samples and the counting of STH eggs under a microscope. Speich and colleagues [7] demonstrated that, independent of the evaluated diagnostic method, the lion's share (~70%) of the total costs of performing egg counts in Zanzibar was made up of salaries. More recently, Leta and colleagues calculated that personnel salaries (~40%) and car rental fees (~50%) made up a combined ~90% of the total study costs when doing a national STH mapping survey in Ethiopia [8]. Hence, the number of samples to be screened, the speed at which technicians can process a single sample, the number of samples that can be processed per day, and thus the number of sampling days, are considered the major cost drivers of programmatic surveys for infection prevalence or therapeutic drug efficacy.

Several different microscopy-based methods (e.g., Kato-Katz thick smear (KK), Mini-FLOTAC, McMaster and FECPAK$^{G2}$) are used to diagnose STH infections in stool, of which some are more complex than others [9–11]. Of all currently applied methods, the WHO-endorsed KK is the most widely established. This method produces smears of 41.7 mg of stool to visualize STH eggs for microscopic identification and counting [12] and is thought to be relatively easy and affordable [7]. The Mini-FLOTAC employs a flotation solution to separate STH eggs from stool debris in a special device prior to counting [9]. The FECPAK$^{G2}$ method is the most recent and innovative diagnostic method [11,13,14]. It is also a flotation-based method, but instead of using a standard microscope, it employs a purpose-made device to accumulate STH eggs in one field of view, and to produce a digital image of this view that can later be marked up by a technician [11]. However, in a previous study, it was shown that both FECPAK$^{G2}$ and Mini-FLOTAC had a clinical sensitivity equal or inferior to a single KK for all STHs [15], and that these flotation-based methods provided lower fecal egg counts (FECs; expressed as eggs per gram of stool (EPG)) compared to KK [16].

Making an evidence-based choice about which FEC method to use in STH control programs remains non-trivial, particularly when a decision-making framework is intended to be applied to a wide range of epidemiological settings. This is because the suitability and the cost of different survey designs and diagnostic techniques will vary by epidemiological setting [17]. For instance, the probability of making correct policy decisions may strongly depend on both the performance of a particular diagnostic method and the associated decision criterion in a particular epidemiological setting [18–23]. For STH, this performance depends on the average intensity of infection in a community as well as the level of variation in egg excretion (between individuals and within individuals over time), and in the case of the evaluation of drug efficacy,

variation in individual drug responses [20,23]. Further, it is important to consider that the total operational cost of a survey will depend on the consumable costs of the diagnostic method used, the survey design (number of samples and number of days spent in the field), and the time needed to count eggs [20]. Importantly, the latter will depend on how many eggs need to be counted, which has not been considered before and which will vary by epidemiological setting and will depend on the goal of the survey (e.g., detecting infection (counting at least one egg) or quantifying intensity of infection (counting all eggs)). Quantifying these costs requires an in-depth analysis of the operational costs of processing samples with different FEC methods.

We aim to provide a general framework for evidence-based recommendations for cost-efficient decision-making in large-scale STH deworming programs based on FEC methods, using monitoring of therapeutic drug efficacy as a case study. To this end, we performed an in-depth analysis of the operational costs to process one sample for three FEC methods (KK, Mini-FLO-TAC and FECPAK[G2]) based on the time-to-result and an itemized cost assessment. Next, we performed a simulation study to determine the probability of correctly concluding that the therapeutic drug efficacy is reduced based on different FEC methods, survey designs and numbers of individuals enrolled, while accounting for the variation in both egg counts and individual drug responses. Finally, we integrated the outcome of the in-depth cost-assessment into the simulation study to determine the most cost-efficient diagnostic test and survey design to detect presence of reduced drug efficacy for the different STH species across different scenarios of STH endemicity.

## Methods

### Ethics statement

Data were collected from four sites during a drug efficacy trial designed to test the equivalence of different FEC methods in attaining estimates of the therapeutic efficacy of a single oral dose of 400 mg albendazole (ALB) against STH infections in school aged children (SAC) [24]. The trial was performed in Brazil, Ethiopia, Lao PDR and Zanzibar (Pemba Island). The study protocol for this trial were reviewed and approved by the Ethics Committee of the Faculty of Medicine and Health Sciences, the University Hospital of Ghent University, Belgium (Ref. No B670201627755; 2016/0266) and the national ethical committees associated with each trial site (Ethical Review Board of Jimma University, Jimma, Ethiopia: RPGC/547/2016; National Ethics Committee for Health Research (NECHR), Vientiane, Lao PDR: 018/NECHR; Zanzibar Medical Research and Ethics Committee, United Republic of Tanzania: ZAMREC/0002/February/ 2015; and the Institutional Review Board from Centro de Pesquisas René Rachou, Belo Horizonte, Brazil: 2.037.205). The trial was retrospectively registered on Clinicaltrials.gov (ID: NCT03465488) on March 7, 2018. Parent(s)/guardians of participants signed an informed consent document indicating that they understood the purpose and procedures of the study, and that they allowed their child to participate. If the child was ≥5 years, he or she had to orally assent in order to participate. Participants of ≥12 years of age were only included if they signed an informed consent document indicating that they understood the purpose and the procedures of the study, and were willing to participate.

### In-depth analysis of the operational costs to process one sample for three FEC methods based on the time-to-result and an itemized cost assessment

**Time-to-result.** Measuring time-to-result for the different FEC methods was part of the drug efficacy trial, which have been extensively described elsewhere [24,25]. During the trial, baseline stool samples were collected from SAC, who were subsequently treated with a single

dose of 400 mg ALB. Between 14 and 21 days after treatment, SAC who were positive for any STH species at baseline were re-sampled to evaluate the reduction in egg output (ERR). At baseline and follow-up, stool samples were processed by duplicate KK (slide A and B), Mini-FLOTAC and FECPAK$^{G2}$ to determine FECs (expressed in EPG) for each STH separately.

Upon arrival in the laboratory, stool samples were first grouped into batches of ten samples (with the remainder in a separate last batch). Subsequently, each individual stool sample was homogenized by stirring with a wooden tongue depressor. Finally, subsamples were taken to be processed according to the different FEC methods. **Fig 1** provides an overview of the different steps timed for each FEC method, including preparing the sample for analysis, counting eggs and data entry (demographic data and FECs), which ultimately resulted in the time-to-result measurement.

Detailed standard operating procedures (SOPs) to time the preparatory steps and the egg counting process are described elsewhere (see S3–S5 Infos of Vlaminck *et al.* [24]); **S1 Info** provides a brief summary. A summary of the SOP to time the data entry is provided in **S2 Info**.

We expressed the time (in seconds) needed to enter data and prepare samples for analysis per batch by dividing the total time recorded per batch by the number of samples within that batch. These calculations included batches gathered at baseline and follow-up. Batches containing fewer than 5 samples were not timed and were excluded from these calculations. We report the average reading time, preparation time and data entry time across batches. The overall mean preparation time per sample was calculated as the mean of batch-specific estimates of time per sample. The data on the timing of egg counting were analyzed at the level of

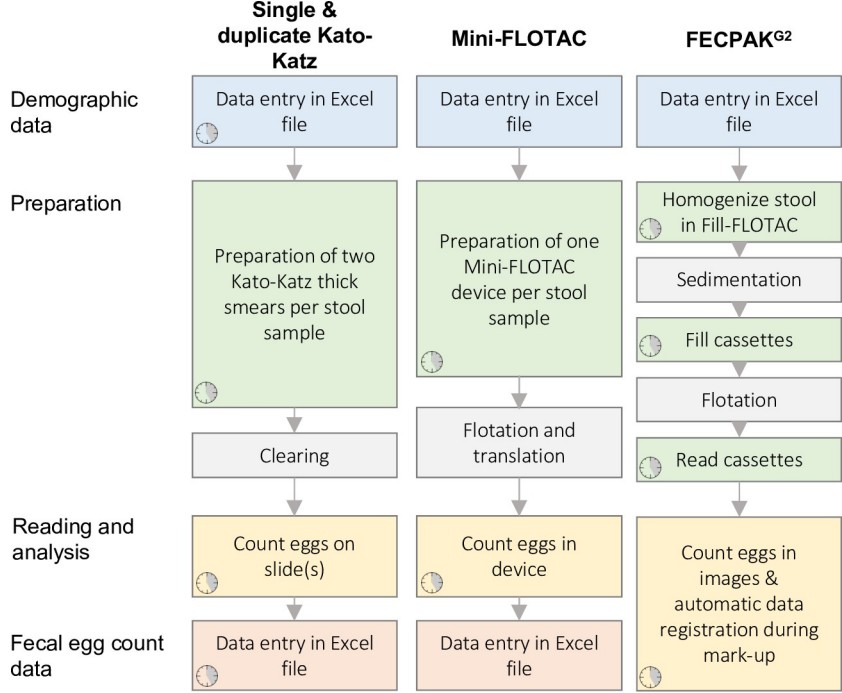

**Fig 1. Overview of the different operational steps for the different FEC methods.** The distinctive steps to perform a Kato-Katz (KK), Mini-FLOTAC or FECPAK$^{G2}$ on a single stool sample are provided in chronological order per method. The procedures are grouped per main subject (**blue**: entry of demographic data; **green**: preparation of the sample; **yellow**: reading of the slide/device or the image to count STH eggs; **red**: entry of fecal egg count data). Waiting steps included in the procedure are indicated in **grey** and represent a fixed amount of time. The small clock symbol indicates what steps have been timed as part of this experiment. Clock clip art from https://openclipart.org/detail/125725/time-temps.

samples. For each FEC method, the correlation between the time required to count and the absolute number of eggs in the sample was quantified using linear regression models. In these models, we predicted the $\log_{10}$-transformed time (in seconds) needed for egg counting (dependent variable) using the square of the $\log_{10}$-transformed total number of STH eggs counted plus 1 as the independent variable. Statistical analyses were conducted in R [26], Microsoft Excel v16.16.7 and Prism version 6.0. for Mac.

**Itemized cost assessment.** We calculated the cost of materials to collect a stool sample in a school setting and the costs to perform the FEC method, including the required equipment, supplies and reagents. For this, we performed an itemized cost assessment considering the cost per unit, the usage over a one-year period and the expected duration of use (in years). A detailed itemized cost assessment to collect stool samples and to perform the FEC method is provided in **S3 Info**. For specific items, such as the KK kit, Mini-FLOTAC or FECPAK$^{G2}$ devices, we used the prices that were either advertised online or obtained through the manufacturer (2020). To estimate the cost of everyday materials, such as scissors, paper, salt and buckets, Ethiopian market prices were used (2020). The cost of a microscope and computer for data-entry were each amortized over 10,000 FEC samples, assuming that they both would be useable for multiple surveys.

## Simulation study to assess the probability of correctly detecting a truly reduced therapeutic efficacy

**Definition of a reduced therapeutic efficacy and survey designs.** For each candidate survey design, we determined the probability that the resulting ERR point-estimate confirmed the presence of reduced efficacy of a single oral dose of 400 mg ALB. Here, we assumed that the true efficacy was 5%-points under the species-specific thresholds specified by WHO (**Table 1**) [5], and we concluded the presence of reduced therapeutic efficacy if the ERR point-estimate was under the WHO threshold. Given that the endemicity at baseline has an impact on the statistical power [27] and the total survey costs [21], we determined the probability of correctly detecting a truly reduced therapeutic efficacy across different scenarios of endemicity.

Currently, it is recommended by WHO to determine the efficacy based on individuals that were egg-positive at baseline only; however, excluding individuals who were egg-negative at baseline from the analysis may result in a substantial overestimation of drug efficacy due to regression to the mean, particularly in low endemic settings or when true drug efficacy is low. Coffeng and colleagues showed previously that this bias could be avoided by a number of alternative survey designs [21]. However, an in-depth analysis of the associated operational costs was missing. Here, we re-evaluate some of the survey designs assessed by Coffeng and colleagues, including the WHO-recommended 'screen and select' design (*SS*; only egg-positive individuals are followed up), the 'screen, select, and retest' (*SSR*; only egg-positive individuals are followed up, but the analysis is based on a second separate baseline stool sample [21]), and the 'no selection' design (*NS*; all enrolled individuals are screened at baseline and follow-up). For the *NS* and *SSR* survey design, we explored two variants, one that was based on a single FEC on the follow-up sample ($NS_{1 \times 1/1x1}$ and $SSR_{1 \times 1/1x1}$), and one that was based on duplicate FECs ($NS_{1 \times 1/1x2}$ and $SSR_{1 \times 1/1x2}$). We did not consider survey designs based on a single FEC on two consecutive stool samples at follow-up FEC ($NS_{1 \times 1/2x1}$ and $SS_{1 \times 1/2x1}$), as sample collection on two consecutive days adds considerable logistical issues while yielding relatively little in terms of precision in drug efficacy estimates [21]. For the WHO-recommended *SS* design we considered two variants: one based on a single FEC both at baseline and follow-up ($SS_{1 \times 1/1x1}$), and one based on duplicate FECs at both time points ($SS_{1 \times 2/1x2}$). The former is recommended in the WHO manual to monitor the therapeutic efficacy of drugs against

**Table 1. Parameterization of the simulation framework for variability in fecal egg counts before and after treatment.**

| Parameter | *Ascaris lumbricoides* | *Trichuris trichiura* | Hookworm |
|---|---|---|---|
| Mean EPG in the population ($\mu$) as measured by KK | | | |
| Endemicity level 1: 1.0–9.9% | 9.6 | 2.8 | 3.7 |
| Endemicity level 2: 10.0–19.9% | 85.2 | 12.9 | 23.7 |
| Endemicity level: 20.0–49.9% | 360.0 | 49.7 | 61.7 |
| Endemicity level: 50.0–100.0% | 2195.5 | 124.7 | 210.3 |
| Variability in EPG between individuals (shape $k_i$) | 0.327 | 0.444 | 0.250 |
| Day-to-day variability in EPG within individuals (shape $k_d$) | 0.510 | 1.000 | 1.000 |
| Weight of stool aliquot ($w_s$; gram) | | | |
| KK | 1/24 | 1/24 | 1/24 |
| Mini-FLOTAC | 1/10 | 1/10 | 1/10 |
| FECPAK$^{G2}$ | 1/34 | 1/34 | 1/34 |
| Relative recovery of eggs from stool ($\rho_s$) | | | |
| KK (reference) | 1.000 | 1.000 | 1.000 |
| Mini-FLOTAC | 0.645 | 1.005 | 0.801 |
| FECPAK$^{G2}$ | 0.248 | 0.152 | 0.569 |
| Variability in egg counts between aliquots based on the same stool sample (shape $k_s$) | | | |
| KK | $\infty^*$ | $\infty^*$ | $\infty^*$ |
| Mini-FLOTAC | 0.579 | 3.022 | 1.465 |
| FECPAK$^{G2}$ | 0.520 | 0.706 | 0.574 |
| Assumed true mean and variability (95%-CI) in drug efficacy (beta distribution with shape parameters $\alpha_{ERR}$ and $\beta_{ERR}$) | 0.80 (0.69–0.89) | 0.35 (0.25–0.45) | 0.75 (0.64–0.84) |
| $\alpha_{ERR}$ | 49.5 | 30.2 | 53.8 |
| $\beta_{ERR}$ | 12.4 | 56.1 | 17.9 |
| WHO definition for reduced therapeutic efficacy based on ERR [5] | <85% | <40% | <80% |

\* Implies $cv_{KK} = 0$, which means that KK-based egg counts from repeated aliquots of the same homogenized stool sample follow a Poisson distribution (no extra over-dispersion in contrast to Mini-FLOTAC or FECPAK$^{G2}$).

schistosomes and STH [5], and the latter is currently being piloted in a number of endemic countries as part of the Starworms project [27].

 **General simulation framework.** For the current simulation study, we adapted the framework described by Coffeng and colleagues [21], accounting for the following sources of variation in egg counts:

1. Inter-individual variability in mean egg intensity due to variation in infection levels between individuals (assumed to follow a gamma distribution);

2. Day-to-day variability in mean egg intensity within an individual due to heterogeneous egg excretion over time (assumed to follow a gamma distribution);

3. Variability in egg counts between repeated aliquots of a stool sample due to the aggregated distribution of eggs in stool (assumed to follow a Poisson or a gamma-Poisson (i.e., negative binomial) distribution);

4. Inter-individual variability in the effect of drug administration in terms of the ERR (assumed to follow a beta distribution).

 For the quantification of each gamma distribution, we followed the approach of Denwood et al [28] in using the coefficient of variation ($cv$) as a standardised measure of variability,

which is related to the shape parameter $k$ of a gamma distribution by taking $k = cv^{-2}$. Species-specific variability between individuals and within individuals over time were estimated based on data from clinical trials during which a duplicate KK was performed on two consecutive stool samples both at baseline and follow-up [29]. STH species and FEC method-specific variability between repeated aliquots of the same stool sample were estimated from the egg count data published by Cools et al. [15]. The average difference between FEC methods in terms of egg recovery performance was also determined as flotation techniques are know to miss unfertilized eggs [30] (see **S4 Info** for details). The parameterization of the simulation framework is summarized in **Table 1**. For a detailed description of the simulation model we refer to **S5 Info**.

Using this framework, we simulated egg counts for all survey designs across four scenarios, each representing different population average baseline FECs. The selection of these scenarios was based on infection levels in the nationwide mapping of STH infections in Ethiopia [31] (see also **S1 Fig**), where each scenario represented the median of school-level mean FEC (in EPG) of one of four endemicity levels (prevalence between 1.0 and 9.9%, 10.0 and 19.9%, 20.0% and 49.9%, or ≥50.0%). For each survey design, we considered a range of 100 to 5,000 individuals (with increments of 5 individuals) that are initially tested at baseline. For each survey design, sample size and endemicity scenario, 10,000 repeated Monte Carlo simulations were performed. In each simulation, the group-based arithmetic mean ERR was calculated using the recommended procedure [32], and the ERR was considered reduced if under the STH species-specific threshold (**Table 1**). For each survey design and sample size, we then calculated the proportion of the 10,000 repeated Monte Carlo simulations that correctly identified therapeutic efficacy as truly reduced ($prob_{reduced}$). If a baseline survey resulted in fewer than 50 egg-positive individuals, the survey was considered to have failed and was discontinued. In those cases, it was considered to not have detected reduced efficacy. In the remainder of the text, the proportion of surveys that fail will be referred to as the "failure rate". All simulations and calculations were performed using the eggsim package [19] in R [26]. This package allows the same calculations to be made for any arbitrary set of parameter values using highly performant C++ code, and is freely available [33].

## Total operational costs to monitor drug efficacy

For each simulated survey, we calculated the total operational costs in terms of (i) the cost of consumables to collect and process samples, (ii) the cost of a single mobile field team comprised of one nurse and three laboratory technicians (including salary and lodging), and (iii) the cost of transport, including car rental, salary of the driver, and gasoline. We assumed that a working day consists of 8 working hours, and that the daily salary of one team was 80 US$ (4 per diems of 22.5 US$) and that the daily cost for transport was 90 US$. Second, we assumed that the team collected samples in the morning (8:00–12:00), and that all collected samples were processed in the afternoon (13:00–17:00). Complete analysis of all samples on the same day implies that the number of samples that can be collected daily is limited, and that this number will vary across FEC methods, phase of the trial (less time for egg counting is required in follow-up samples) and endemicity (more time for egg counting is required in highly endemic areas). Note that we do not consider costs for the establishment and maintenance of laboratory infrastructure. We further assumed that all work takes place on regular working days, that the team does not take any breaks during processing, and that all samples are collected from a single school/community without loss to follow-up. All cost calculations were based on the itemized cost-assessment described above. Technical details on how the total costs were calculated can be found in **S6 Info.**

## Results

### Time-to-result

During the drug efficacy trial surveys performed in Brazil, Ethiopia, Lao PDR, and Tanzania, we assessed the mean time-to-result (i) to prepare stool samples ($T_{prep,X}$, $X$ representing the number of aliquots per sample), (ii) to count eggs, (iii) to digitize demographic data ($T_{demography}$), and (iv) to digitize the FEC results ($T_{record,X}$). The time analysis is illustrated in **Fig 2**. Overall, a duplicate KK consumed the most time, requiring on average (standard deviation) 989 sec (449). A single KK consumed the least amount of time and required on average 507 sec (318). The mean time-to-result for a single Mini-FLOTAC or FECPAK$^{G2}$ method were 786 sec (513) and 802 sec (329), respectively. The percentage of time-to-result spent on the egg counting process was approximately 80% for both KK (single KK: 413 sec out of 507sec; duplicate KK: 820 sec out of 989 sec) and Mini-FLOTAC (632 sec out of 786 sec), while this was 23% for FECPAK$^{G2}$ (185 sec out of 802 sec). For the latter method, most of the time-to-result (74%) was spent preparing the samples for analysis (596 sec out of 802 sec).

As expected, counting a larger number of STH eggs required more reading time, where the log transformed total time required to count all eggs could be well described as a linear function of the square of the base-10 log-transformed total egg counts (**Fig 3**). **Table 2** summarizes the average time required for the different steps included in the total survey costs for each FEC method separately. To obtain an estimate for a single KK preparation we divided the average time to prepare a duplicate KK (135 sec) by two (= 67 sec). As a second Mini-FLOTAC can be filled from the same Fill-FLOTAC (no need to weigh and homogenize the sample for a duplicate Mini-FLOTAC), we multiplied the mean time required to process a single Mini-FLOTAC (131 sec) by 1.5 to estimate the time for a duplicate Mini-FLOTAC (197 sec). To estimate the time for a duplicate FECPAK$^{G2}$, we doubled the time for each of the different preparatory steps, except for the step to prepare the samples in the FECPAK$^{G2}$ sedimenters, resulting in a total time of 1,050 sec (= 142 sec + 2 x 174 sec + 2 x 280 sec). Similarly, the mean time needed to enter one duplicate KK result was 18 sec; the time needed to record FEC results based on single KK and Mini-FLOTAC was assumed to be half that value (9 sec). For the FECPAK$^{G2}$ method, no FEC data entry was required as the software automatically registers and stores mark-up data. **S7 Info** provides more detailed information (number of batches timed; the average units per batch; average and SD time) on each step of the sample analysis process, starting with the timing of the demographic data entry followed by the preparation phase, the egg counting process, and the time it took to enter FEC data.

### Itemized cost assessment

The costs associated with the materials for stool sample collection in schools and to process stool samples for a single or duplicate KK, mini-FLOTAC and FECPAK$^{G2}$ are reported in detail in **Table 3**. In summary, the costs associated with sampling a single sample ($cost_{sample}$) was US\$ 0.57. The material costs to perform a single FEC ($cost_{aliquot,1}$) were US\$ 1.37 for KK, US\$ 1.51 for the mini-FLOTAC method, and US\$ 1.69 for the FECPACK$^{G2}$ method. When a duplicate FEC was performed on the same sample the material costs ($cost_{aliquot,2}$) were US\$ 1.51 (KK), US\$ 1.87 (Mini-FLOTAC), and US\$ 2.73 (FECPAK$^{G2}$).

### Failure rate, probability of correctly detecting reduced therapeutic efficacy and the corresponding total survey costs

Given the large number of possible scenarios (981 sample sizes x 6 study designs x 4 levels of endemicity x 3 FEC methods x 3 STH species = 211,896 scenarios), and the different output

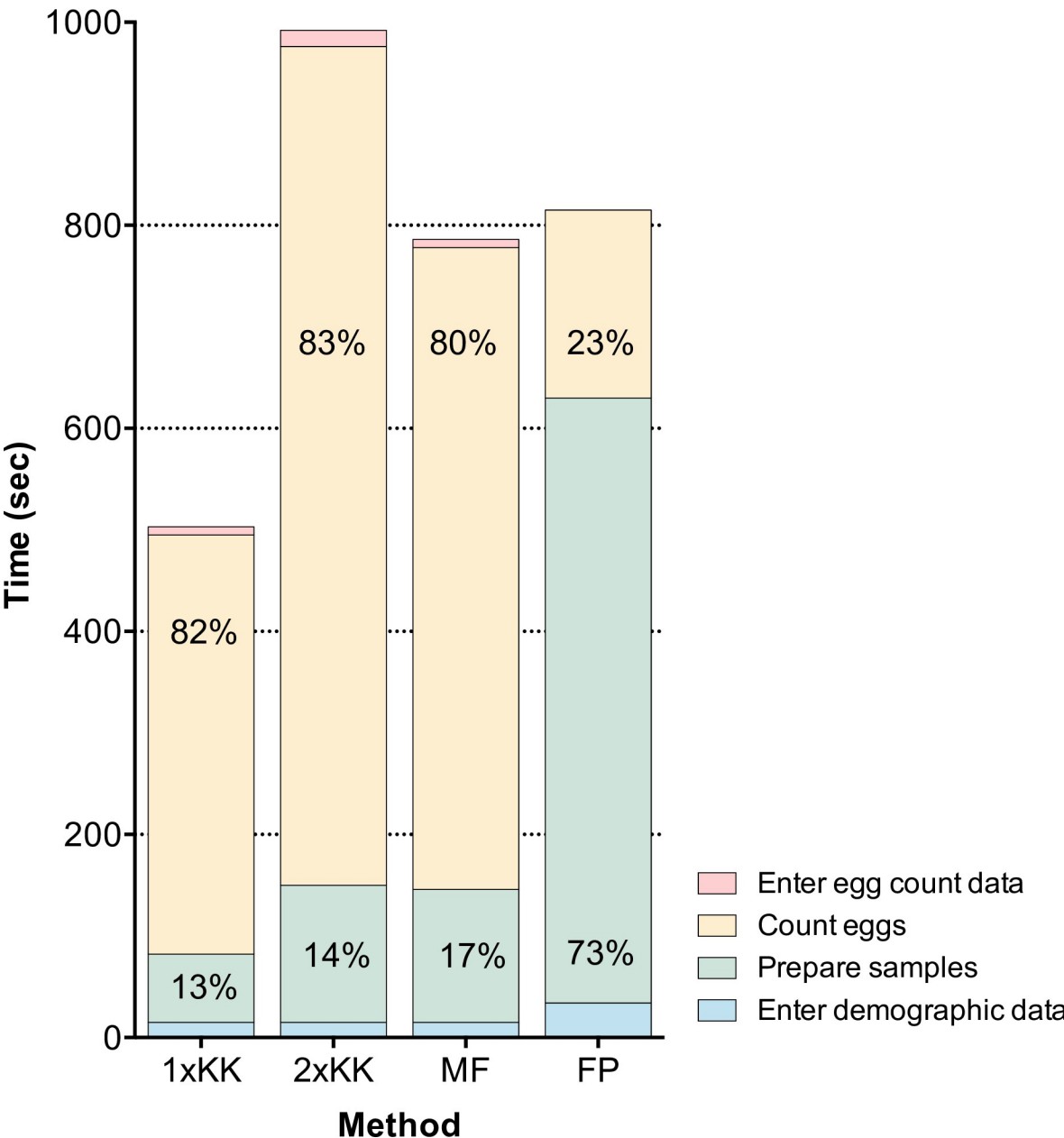

**Fig 2. Time required to quantify soil-transmitted helminth infections in stool by four fecal egg count methods.** The height of the bars represents the mean time (in sec) needed to enter demographic data (**blue**), to perform the preparation phase (**green**), to count eggs (**yellow**) and to enter egg count data (**red**) for a single (1xKK) and duplicate Kao-Katz (2xKK), Mini-FLOTAC (MF) and FECPAK^G2 (FP). The relative proportion (in %) of total time required to perform the preparation phase and to count is reported inside the bars.

parameters (failure rate, $prob_{reduced}$ and $cost_{total}$), we first illustrate the performance of only KK-based survey designs in areas that are low endemic for hookworm(mean FEC = 3.7 EPG) (**Fig 4**). **Fig 4A** shows that the failure rate, i.e., the risk of observing fewer than 50 egg-positive individuals at baseline, is high ($> 25\%$) when $< 250$ subjects were enrolled. For sample sizes of about 250 to 750 subjects, the failure rate was lower for a $SS_{1 \times 2/1x2}$ survey design compared to the other survey designs, as duplicate KK results in higher sensitivity for detecting at least one egg in the baseline samples. To reduce the failure rate to $< 1\%$, at least 440 individuals needed

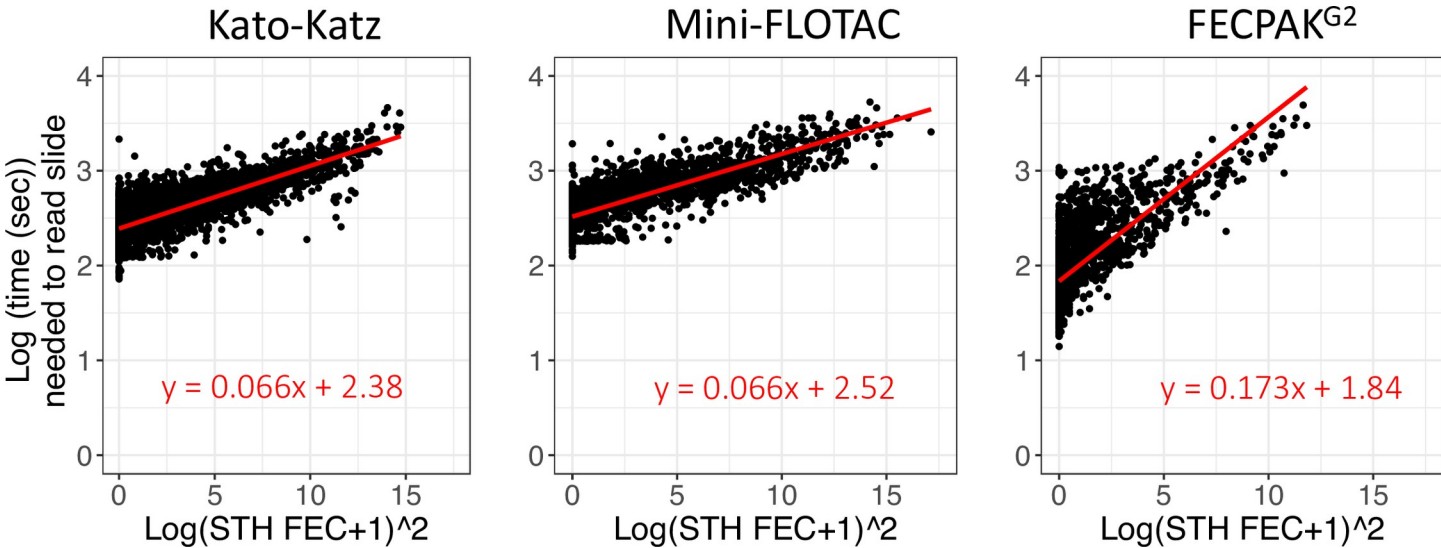

**Fig 3. The reading time as a function of the number of STH eggs counted in a sample.** This figure represents the reading time as a function of the number of STH eggs counted in a sample for single Kato-Katz (KK), Mini-FLOTAC and FECPAK[G2] separately. All egg counts represent raw egg counts (not in eggs per gram of stool). The red line represents the linear regression line. The function of the regression line is also provided.

to be enrolled for a $SS_{1 \times 2/1x2}$, while this was at least 690 for the other survey designs. For the $NS$ and $SSR$ survey designs, the probability of correctly detecting reduced drug efficacy ($prob_{reduced}$) increased with the number of individuals enrolled (**Fig 4B**) but varied between these survey designs. For example, when 700 individuals were recruited the $prob_{reduced}$ equalled 84% for $NS_{1 \times 1/1x2}$, 79% for $NS_{1 \times 1/1x1}$, 71% for $SSR_{1 \times 1/1x2}$ and 67% for $SSR_{1 \times 1/1x1}$. For $SS$ surveys, $prob_{reduced}$ did not increase beyond 6% and decreased again with increasing sample sizes over 400, which is driven by the increasingly precise (due to sample size) but systematically overestimated drug efficacy (due to regression towards the mean).

For a given sample size, the most expensive survey designs were , $NS_{1 \times 1/1x2}$, $NS_{1 \times 1/1x1}$ and $SS_{1 \times 2/1x2}$, while the two $SSR$ designs and $SS_{1 \times 1/1x1}$ were the cheapest (**Fig 4C**). For example, when enrolling 700 individuals, the mean total survey cost ($cost_{total}$) was at least 4,000 US$ for $NS_{1 \times 1/1x2}$, $NS_{1 \times 1/1x1}$ and $SS_{1 \times 2/1x2}$, while for the two $SSR$ designs and $SS_{1 \times 1/1x1}$ the mean $cost_{total}$ was around 3,000 US$ or less. To determine the most cost-efficient survey design, we plotted the probability of detecting reduced efficacy ($prob_{reduced}$) against total survey cost ($cost_{total}$) (**Fig 4D**). For survey budgets up to 2,600 US$, the two $SSR$ survey designs maximized the

**Table 2. Overview of parameters that determine the time required to process a single stool sample.**

| Symbol | Description | Average time required (seconds) | | |
|---|---|---|---|---|
| | | KK | Mini-FLOTAC | FECPAK[G2] |
| $T_{demography}$ | Time to enter demographic data | 15 | 15 | 34 |
| $T_{prep,X}$ | Time to prepare a stool sample | | | |
| $X = 1$ | One aliquot | 67 | 131 | 596 |
| $X = 2$ | Two aliquots | 135 | 197 | 1,050 |
| $f(c)$ | Time required to count $c$ eggs (not EPG) in a single aliquot | $10^{\wedge}(2.3896 + 0.0661 \times \log_{10}(c+1)^2)$ | $10^{\wedge}(2.5154 + 0.0661 \times \log_{10}(c+1)^2)$ | $10^{\wedge}(1.8349 + 0.1731 \times \log_{10}(c+1)^2)$ |
| $T_{record,X}$ | Time to record count data | | | |
| $X = 1$ | One aliquot | 9 | 9 | 0 |
| $X = 2$ | Two aliquots | 18 | 18 | 0 |

**Table 3. Overview of cost per unit of consumables, salary and travel.**

| Symbol | Description | Cost (US$) | | |
|---|---|---|---|---|
| | | KK | Mini-FLOTAC | FECPAK$^{G2}$ |
| $cost_{sample}$ | Cost of collecting a single stool sample | 0.57 | 0.57 | 0.57 |
| $cost_{aliquot,X}$ | Cost per aliquot when preparing $X$ aliquots from the same stool sample | | | |
| $X = 1$ | One aliquot | 1.37 | 1.51 | 1.69 |
| $X = 2$ | Two aliquots | 1.51 | 1.87 | 2.73 |
| $salary_{perdiem}$ | Daily salary for every technician and nurse on the mobile Team | 22.50 | 22.50 | 22.50 |
| $travel_{perdiem}$ | Daily cost of car rental, including petrol and driver wages | 90.00 | 90.00 | 90.00 |

probability to detect reduced drug efficacy. For budgets between 2,600 and 4,200 US$, the $SSR_{1 \times 1/1x2}$ design was the most cost-efficient. For budgets between 4,200 and 5,200 US$, $NS_{1 \times 1/1x1}$ resulted in the highest $prob_{reduced}$, whereas for a budget of 5,200 US$ or more, $NS_{1 \times 1/1x2}$ was the most cost-efficient survey design. To reduce the risk of falsely concluding adequate drug efficacy to <20% ($prob_{reduced} \geq 80\%$), $NS_{1 \times 1/1x1}$ was the most cost-efficient option (red line; $cost_{total}$ = 5,000 US$), with $NS_{1 \times 1/1x2}$ as a close runner-up (beige line; $cost_{total}$ = 5,200 US$).

Second, we explored the impact of the different FEC methods across the six survey designs for hookworms in the same endemicity level as above (**Fig 5**). Generally, deploying Mini-FLOTAC and FECPAK$^{G2}$ did not greatly improve the $prob_{reduced}$ for *SS* survey designs. For the other survey designs, Mini-FLOTAC and KK achieved were equally cost-efficient (lines are close to each other). For Mini-FLOTAC, the cheapest survey design to obtain a $prob_{reduced} \geq$ 80% was an $NS_{1 \times 1/1x2}$ survey based on 495 individuals, at a cost of 5,246 US$ (**S6 Fig**). For KK, this was $NS_{1 \times 1/1x1}$ based on 730 individuals at a cost of 4,987 US$ (**Fig 4**). For FECPAK$^{G2}$, the $prob_{reduced}$ remained below 85.2%, even when both sample size (2,000) and available budget (27,140 US$) were maximized (see **S7 Fig** for details on the impact of sample size).

When determining the most cost-efficient survey design for the other two STH species at low endemicity level (*A. lumbricoides*: mean FEC = 9.6 EPG, **S2 and S3 Figs**; *T. trichiura*: mean FEC = 2.8 EPG, **S4 and S5 Figs**), we noted three important differences compared to hookworm. First, the risk for a failed survey was remarkably lower for *A. lumbricoides*. While the failure rate is 8.5% when 250 subjects are enrolled for a survey ($SS_{1 \times 2/1x2}$ targeting *A. lumbricoides* (**S2A Fig**), this was 98.7% and 97.4% for *T. trichiura* (**S4A Fig**) and hookworms (**Fig 4A**), respectively. As a consequence of this, the mean $cost_{total}$ and the sample size at which $prob_{reduced} \geq 80\%$ was lower compared to the other STHs (*A. lumbricoides*: **S2D Fig**: $NS_{1 \times 1/1x1}$ at mean $cost_{total}$ = 2,522 US$; *T. trichiura*: **S4D Fig**: $NS_{1 \times 1/1x2}$ at mean $cost_{total}$ = 19,628 US$; Hookworm: **Fig 4D**: $NS_{1 \times 1/1x1}$ at mean $cost_{total}$ = 4,987 US$). Second, in contrast to hookworms, where $prob_{reduced}$ for a given budget differed only marginally between Mini-FLOTAC and Kato-Katz thick smear (**Fig 5**), the differences FEC methods were more substantial for *A. lumbricoides* and *T. trichiura*). For *A. lumbricoides* (**S3 Fig**), KK provided the highest $prob_{reduced}$ for the same budget, while this was Mini-FLOTAC for *T. trichiura* (**S5 Fig S5**). Third, none of the survey designs achieved a $prob_{reduced} \geq 80\%$ for *T. trichiura*, given the maximum simulated sample size of 2,000 individuals (**S4B Fig**).

In **Fig 6**, we show the impact of pre-treatment endemicity on the probability of correctly identifying reduced drug efficacy based on KK. When surveys were conducted in higher levels of endemicity, a higher $prob_{reduced}$ was obtained for the same budget. Although this was observed for all three STH species and all six survey designs, this increase was most distinct for *SS* survey designs. This was to be expected as the bias due to regression towards the mean in *SS* survey designs is known to decrease with higher infection levels. Although *SS* survey designs

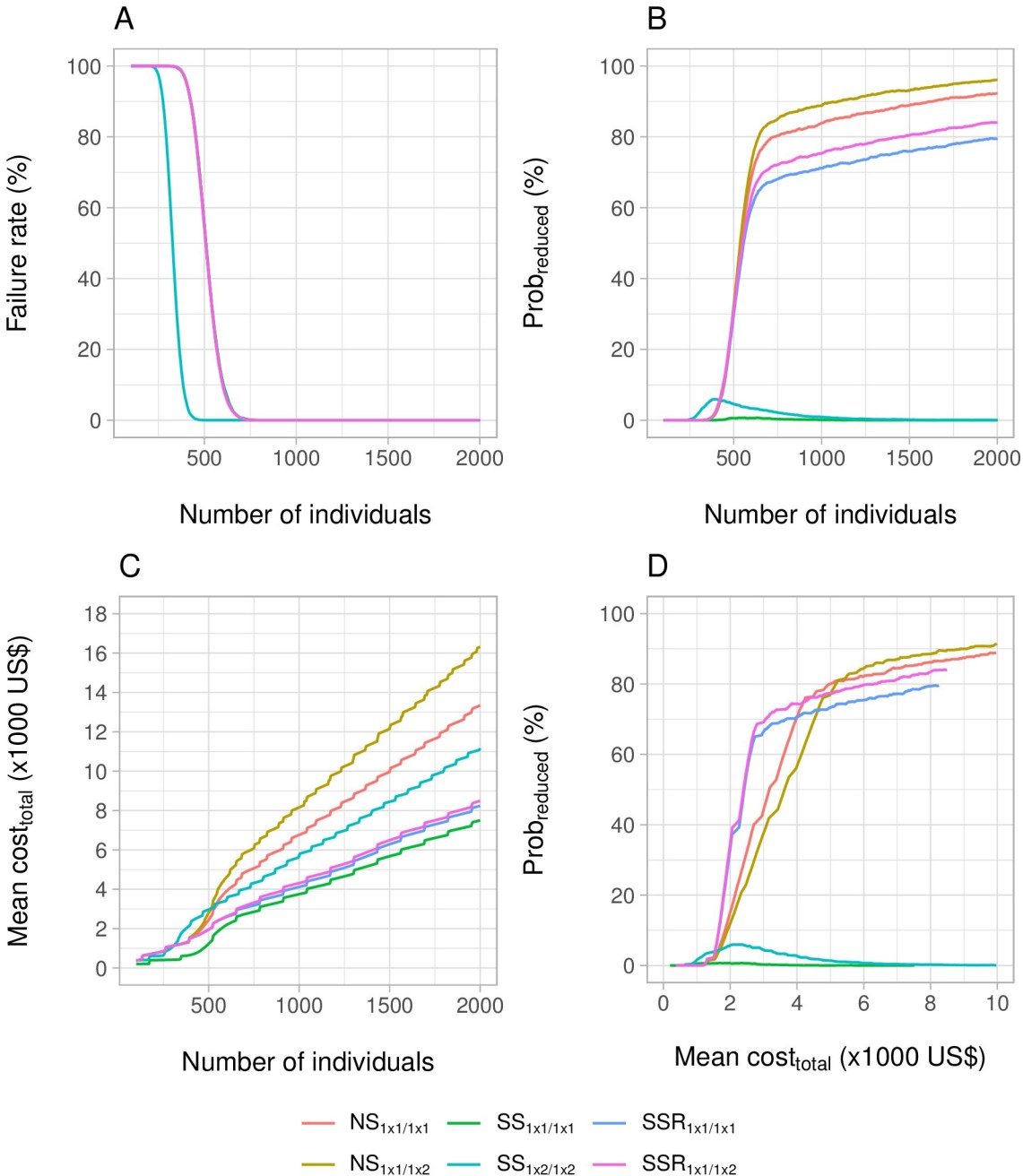

**Fig 4. The failure rate, the probability of correctly concluding reduced drug efficacy and the total survey cost across six survey designs.** This figure shows the impact of the survey design and sample size on the failure rate (**Panel A**), probability of correctly detecting truly reduced efficacy ($prob_{reduced}$; **Panel B**) and the mean total survey cost ($cost_{total}$; **Panel C**). To gain more insights into the most cost-efficient survey design, the probability of correctly detecting reduced drug efficacy $prob_{reduced}$ was plotted as a function of the mean $cost_{total}$ (**Panel D**). For each of the four panels, we only consider the use of Kato-Katz in areas with low levels of hookworm infection (mean FEC = 3.7 EPG). NS = no selection; SS = screen and select; SSR = screen, select, and retest. Note, for panel A, all survey designs other than SS1x2/1x2 are identical to SSR1x1/1x2.

rarely resulted in a correct detection of a truly reduced therapeutic drug efficacy when endemicity levels were low (top row panels **Fig 6**), they almost reach the highest $prob_{reduced}$ at a cost of the cheapest survey design at the highest levels of endemicity for both *A. lumbricoides* and hookworms (bottom row panels **Fig 6**). For *T. trichiura*, the performance of *SS* survey

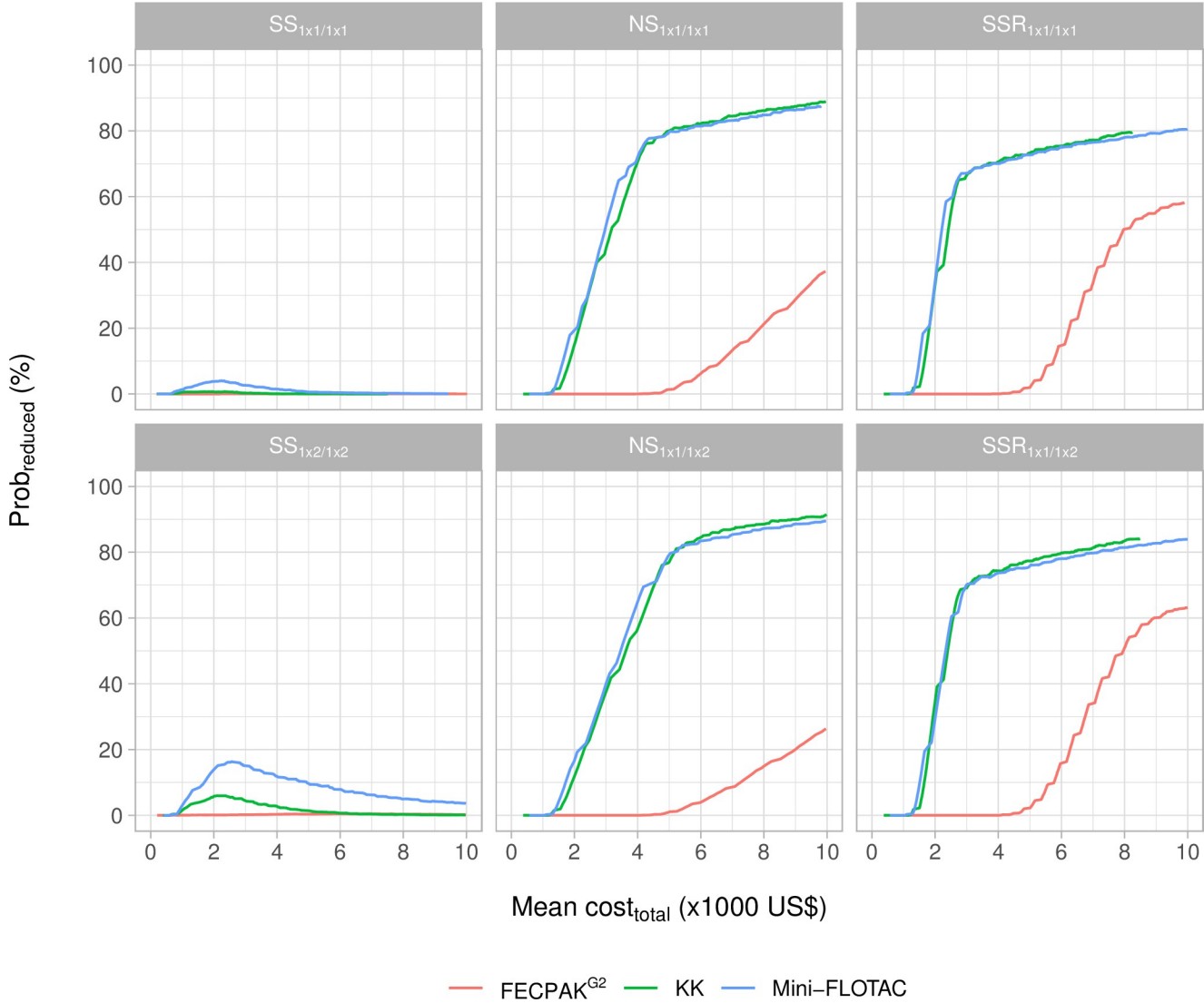

**Fig 5. The probability of correctly detecting presence of reduced drug efficacy and the total survey cost for three FEC methods across six survey designs.** This figure plots the probability of correctly identifying reduced therapeutic efficacy ($prob_{reduced}$) as a function of the mean total survey costs ($cost_{total}$) for the three different FEC methods (Kato-Katz thick smear (KK), Mini-FLOTAC and FECPAK$^{G2}$; colored lines) and six survey designs (different panels). For each panel, we only consider areas that are low endemic for hookworm (mean FEC = 3.7 EPG). NS = no selection; SS = screen and select; SSR = screen, select, and retest.

designs remained relatively poor, which is logical as the regression towards the mean is expected to be higher when true drug efficacy is lower (45% for *T. trichiura* in the simulations). This figure also highlights a shift in the most cost-efficient survey design: while at low level of endemicity, the most cost-efficient survey design depends on the available funds, for higher endemicities, only the $NS_{1 \times 1/1x1}$ survey design maximizes the $prob_{reduced}$ for any available budget.

In **Table 4**, we provide the sample size and mean $cost_{total}$ for those survey designs that detected reduced efficacy with $prob_{reduced} \sim 80\%$ at the lowest cost for each of the different STH species. Generally, this table confirms that the $NS_{1 \times 1/1x1}$ survey design in combination with KK was the most cost-efficient choice to assess therapeutic drug efficacy in all scenarios of STH species and endemicity. Only when surveys were conducted in areas where endemicity

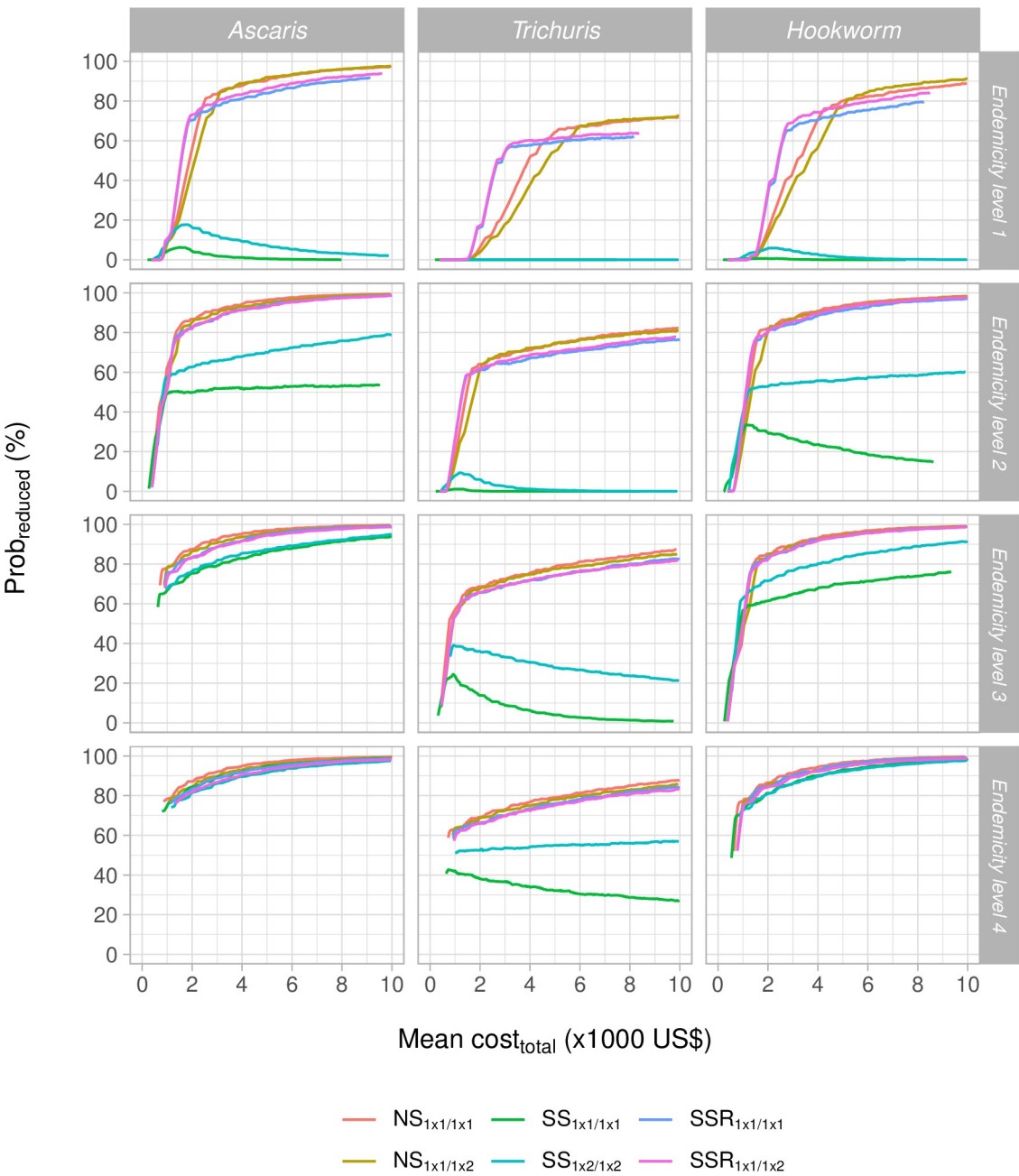

**Fig 6. The probability of correctly detecting presence of reduced drug efficacy and the total survey cost for six survey designs across four levels of endemicity when deploying Kato-Katz.** This figure plots the probability of correctly identifying reduced therapeutic efficacy ($prob_{reduced}$) as a function of the mean total survey costs ($cost_{total}$) across six survey designs for the three soil-transmitted helminth species and four levels of endemicity (see **Table 1**).

of *T. trichiura* infections were low, the $NS_{1 \times 1/1x2}$ survey combined with Mini-FLOTAC was more cost-efficient.

## Discussion

In this paper, we present new evidence-based recommendations for cost-efficient monitoring of therapeutic drug efficacy against STH, using a simulation framework that captures

important interactions between STH epidemiology (variability in egg counts due to various sources), diagnostic test performance (species- and method-dependent egg recovery and count variability), survey design (bias and accuracy that change with endemicity) and operational costs (which change with endemicity, diagnostic method and survey design). With this framework, we address the challenge of minimizing operational costs of STH monitoring in resource-limited settings without jeopardizing the quality of the decision-making. We performed an in-depth analysis of the operational costs to process one sample for three FEC methods (KK, Mini-FLOTAC and FECPAK$^{G2}$) based on the time-to-result and an itemized cost assessment. Next, we simulated how the probability of correctly detecting reduced therapeutic drug efficacy depends on different FEC methods and survey designs, accounting for sources of variation in egg counts as quantified based on several STH datasets. Finally, we integrated the outcome of the in-depth cost-assessment into the simulation study and determined the most cost-efficient survey design to detect presence of reduced drug efficacy across different scenarios of STH endemicity. Overall, we confirm that KK is the best FEC method to monitor therapeutic drug efficacy, but that the survey design currently recommended by WHO should be updated.

## Single KK is the cheapest and least time-consuming method

The mean time and cost for material to process one sample varied from ~8.5 min (single KK) to ~16.5 min (duplicate KK), and from US$ 1.37 (single KK) to US$ 1.69 (FECPAK$^{G2}$). Our study found that a single KK is both the cheapest and least-time consuming of the three FEC methods evaluated. Although a comparison across studies is not straightforward, as differences in laboratory time can be explained by differences in endemicity (laboratory time varies significantly with the number of eggs counted, as we show here) and possibly also the level of expertise of laboratory technicians, other researchers generally reported a similar laboratory time for both single ([35: ~9.5 min; [34]: ~11.0 min; [8]: ~5.0 min) and duplicate KK ([7]: ~16.6 min). The cost for material estimated in the current study (single KK: US$ 1.95; duplicate KK: US$ 2.17), were higher than those reported by Speich *et al.* [7] (single KK: US$ 0.03; duplicate KK: US$ 0.04). These differences can be explained by the fact that we included fixed survey costs (0.60 US$, e.g., gloves and permanent markers). In addition, while Speich *et al* [7] re-used the templates for 50 samples, we opted for single use of materials as there was only limited mesh and cellophane in one kit. For the other two FEC methods, the laboratory time and cost for material were ~13.1 min and US$ 1.51 for Mini-FLOTAC, and ~13.5 min and US$ 1.69 for FECPAK$^{G2}$. However, data on laboratory time and cost for material to compare our results are either scarce (Mini-FLOTAC: 8–12 min [9]) or absent (FECPAK$^{G2}$). It is also important to note that our estimates of laboratory time and material costs did not include the washing of devices, and that we based our costs based on an Ethiopian market in 2020.

## Revision of the WHO guidelines to monitor drug efficacy is warranted

WHO currently recommends a selection and screen approach during which a single stool sample is processed by a single KK both at baseline and follow-up [5]. Although our study confirms that KK is the FEC method of choice, it indicates that the recommended survey design will often result in poor decision-making due to overestimation of drug efficacy (because of regression towards the mean) at a relatively high cost. Instead, a KK-based survey design where all children are followed up regardless of their baseline infection status (the "no selection" or *NS* design) should be preferred as it yields unbiased results at the lowest operational cost. The "screen, select, and retest" strategy, where individuals who are egg-positive at baseline are retested based on a new pre-treatment stool sample (the *SSR* design) [21], was found

**Table 4. The most cost-efficient survey design and FEC method to monitor the therapeutic drug efficacy against STHs.**

| STH species | Mean FEC (EPG) | Survey design | FEC method | Sample size | Mean number of days | Mean $cost_{total}$ (US$) |
|---|---|---|---|---|---|---|
| *A. lumbricoides* | | | | | | |
| | 9.6 | $NS_{1 \times 1/1x1}$ | Kato-Katz | 380 | 6 | 2,522 |
| | 85.2 | $NS_{1 \times 1/1x1}$ | Kato-Katz | 165 | 4 | 1,358 |
| | 360.0 | $NS_{1 \times 1/1x1}$ | Kato-Katz | 145 | 4 | 1,282 |
| | 2195.5 | $NS_{1 \times 1/1x1}$ | Kato-Katz | 135 | 4 | 1,243 |
| *T. trichiura* | | | | | | |
| | 2.8 | $NS_{1 \times 1/1x2}$ | Mini-FLOTAC | 1,680 | 54 | 17,313 |
| | 12.9 | $NS_{1 \times 1/1x1}$ | Kato-Katz | 1,215 | 19 | 8,134 |
| | 49.7 | $NS_{1 \times 1/1x1}$ | Kato-Katz | 830 | 14 | 5,740 |
| | 124.7 | $NS_{1 \times 1/1x1}$ | Kato-Katz | 790 | 14 | 5,523 |
| Hookworms | | | | | | |
| | 3.7 | $NS_{1 \times 1/1x1}$ | Kato-Katz | 730 | 12 | 4,987 |
| | 23.7 | $NS_{1 \times 1/1x1}$ | Kato-Katz | 245 | 4 | 1,666 |
| | 61.7 | $NS_{1 \times 1/1x1}$ | Kato-Katz | 185 | 4 | 1,437 |
| | 210.3 | $NS_{1 \times 1/1x1}$ | Kato-Katz | 160 | 4 | 1,340 |

to be somewhat less cost-efficient than the *NS* design. However, as previously discussed [21], because the *SSR* design will yield more egg-positive individuals than a *NS* design based on the same budget, *SSR* could still be considered for study objectives that require a minimal number of eggs or egg-positive individuals, such as genotyping to identify resistance-conferring polymorphisms.

## Identifying the most cost-efficient study design for any programmatic survey

In the present study, the laboratory time and cost analysis were used to identify the most cost-efficient design for monitoring drug efficacy, but these analyses can also be used to identify the most cost-effective study design for any other type of survey. Although this concept is not new and has been applied in the past [35,36], the level of detail that we present for each of the different FEC allows for fine-tuned cost-efficiency analysis of any programmatic survey. For instance, this framework would also lend itself well to assess the cost and performance of surveys for decisions about stopping or scaling down preventive chemotherapy against STH and/or assess the potential value and cost-efficiency of (new) diagnostic techniques with different (hopefully better) performance and throughput than FEC methods, but potentially higher reagent costs [37].

## Automated egg counting would further reduce operational costs

As highlighted by the present study, egg counting is the most time-consuming step for KK and Mini-FLOTAC (80%). An obvious cost-saving strategy that would further reduce the operational costs is automated egg counting using a scanning/imaging device and artificial intelligence-based egg-recognition software to identify and report egg counts. A variety of artificial intelligence based digital pathology (AI-DP) devices are currently being studied [38–42]. However, a complete AI-DP device is currently not commercially available, despite the successful examples for other parasitic infections (malaria: CellsCheck, http://www.biosynex.com; Loa Loa: [43]). At the time of writing, FECPAK$^{G2}$ was probably the most advanced, but automated egg-recognition on the created images by existing STH egg-recognition software has proven

difficult or impossible (S5 Info of Cools et al. [15]). In addition, our study highlighted that due to its poor diagnostic performance [15], FECPAK[G2] is not recommended to monitor therapeutic drug efficacy in STH control programs. Despite these challenges, there are ongoing investments around each of the FEC methods to progress towards a complete point-of-care platform with automated egg counting and built-in data analysis [39,42].

## Strengths and limitations

This is the first comprehensive study that compares the operational costs between the most-used FEC methods in STH surveys. It is important to note that the estimated costs are institute- and context-specific, and hence the reported values should not be interpreted as absolute. However, a major strength of our framework is that assumptions about costs can be easily adapted to represent particular settings. Because our framework aims to compare different survey designs and diagnostic methods, we do not consider costs that can be reasonably assumed to be the similar across different survey designs and different FEC methods: salary for senior staff to supervise the field activities, report, and analyze the data; power supplies; laboratory rent; per diems for days when work is not possible (e.g., weekends); time required for the field team to travel to and return from the study location at the start and end of the survey; and time required to set up and clean laboratories and inform schools and local health authorities prior to surveys. In addition, we do not consider the time required to travel to new another study location if the target sample size cannot be reached in single site, meaning that some of the larger recommended survey designs (e.g., for *T. trichiura*) are potentially somewhat more expensive than we estimated. A possibly relevant simplifying assumption that we made is that survey teams work constantly without any break. This may have led to a slight overestimation of the performance per cost (**Fig 5**) for each of the different FEC methods (more breaks because of manual egg counting) compared to FECKPAK[G2]. Finally, it is important to highlight that each of the trials were conducted by well-trained teams (and hence the laboratory time for a less experienced team might be underestimated), and that we assumed that no individuals would be lost to follow-up. Theoretically, each of the aforementioned factors could be included in our simulation framework, although we do not expect that the presented relative rankings of FEC methods and survey designs would be affected by the inclusion of this additional real-life complexity. However, some of these aspects, like laboratory infrastructure, will have to be considered when comparing FEC methods with other diagnostic techniques such as quantitative polymerase chain reaction.

## Conclusion

We confirm that Kato-Katz is the FEC method of choice for assessing drug efficacy, but that the current WHO-recommended screen and select survey design should be replaced by a no-screen survey design. Our detailed analysis of laboratory time and material costs allows for a cost-efficiency assessment of other FEC-based programmatic decisions as well, like decisions regarding stopping and scaling down of preventive chemotherapy. In addition, the flexible and holistic design of our simulation framework allows further study of alternative diagnostic techniques that aim to further decrease operational costs while maintaining or improving the quality of decision-making, like automated egg counting.

## Supporting information

**S1 Info. Summary of the standard operating procedures to time the preparatory steps and the egg counting for three FEC methods.**
(PDF)

**S2 Info. Summary of the standard operating procedures to time the data entry.**
(PDF)

**S3 Info. A detailed itemized cost assessment both to collect and process stool samples and for each FEC method.**
(XLSX)

**S4 Info. Methods to estimate the FECs of Mini-FLOTAC and FECPAK$^{G2}$ based on duplicate Kato-Katz.**
(PDF)

**S5 Info. Detailed description and parameterization of the simulation model.**
(PDF)

**S6 Info. The calculation of the total operational costs to monitor drug efficacy.**
(PDF)

**S7 Info. Detail information on the calculation of the time-to-result.**
(PDF)

**S1 Fig. The range of school mean fecal egg counts for four levels of school prevalence for three soil-transmitted helminths.**
(PDF)

**S2 Fig. The failure rate, the probability of correctly concluding reduced drug efficacy and the total survey cost across six survey designs for *Ascaris*.** This figure shows the impact of the survey design and sample size on the failure rate (**Panel A**), probability of correctly detecting truly reduced efficacy ($prob_{reduced}$; **Panel B**) and the mean total survey cost ($cost_{total}$; **Panel C**). To gain more insights into the most cost-efficient survey design, the probability of correctly detecting reduced drug efficacy $prob_{reduced}$ was plotted as a function of the mean $cost_{total}$ (**Panel D**). For each of the four panels, we only consider the use of Kato-Katz in areas with low levels of *Ascaris* infection (mean FEC = 9.6 EPG). NS = no selection; SS = screen and select; SSR = screen, select, and retest. Note, for panel A, all survey designs other than SS1x2/1x2 are identical to SSR1x1/1x2.
(PDF)

**S3 Fig. The probability of correctly detecting presence of reduced drug efficacy and the total survey cost for three FEC methods across six survey designs for *Ascaris*.** This figure plots the probability of correctly identifying reduced therapeutic efficacy ($prob_{reduced}$) against *Ascaris* infections as a function of the mean total survey costs ($cost_{total}$) for the three different FEC methods (Kato-Katz thick smear (KK), Mini-FLOTAC and FECPAK$^{G2}$; colored lines) and six survey designs (different panels). For each panel, we only consider areas that are low endemic for *Ascaris* (mean FEC = 9.6 EPG). NS = no selection; SS = screen and select; SSR = screen, select, and retest.
(PDF)

**S4 Fig. The failure rate, the probability of correctly concluding reduced drug efficacy and the total survey cost across six survey designs for *Trichuris*.** This figure shows the impact of the survey design and sample size on the failure rate (**Panel A**), probability of correctly detecting truly reduced efficacy ($prob_{reduced}$; **Panel B**) and the mean total survey cost ($cost_{total}$; **Panel C**). To gain more insights into the most cost-efficient survey design, the probability of correctly detecting reduced drug efficacy $prob_{reduced}$ was plotted as a function of the mean $cost_{total}$

(**Panel D**). For each of the four panels, we only consider the use of Kato-Katz in areas with low levels of *Trichuris* infection (mean FEC = 2.8 EPG). NS = no selection; SS = screen and select; SSR = screen, select, and retest. Note, for panel A, all survey designs other than SS1x2/1x2 are identical to SSR1x1/1x2.
(PDF)

**S5 Fig. The probability of correctly detecting presence of reduced drug efficacy and the total survey cost for three FEC methods across six survey designs for *Trichuris*.** This figure plots the probability of correctly identifying reduced therapeutic efficacy ($prob_{reduced}$) against *Trichuris* infections as a function of the mean total survey costs ($cost_{total}$) for the three different FEC methods (Kato-Katz thick smear (KK), Mini-FLOTAC and FECPAK$^{G2}$; colored lines) and six survey designs (different panels). For each panel, we only consider areas that are low endemic for *Trichuris* (mean FEC = 2.8 EPG). NS = no selection; SS = screen and select; SSR = screen, select, and retest.
(PDF)

**S6 Fig. The probability of correctly detecting presence of reduced drug efficacy and the total survey cost for six survey designs across four levels of endemicity when deploying Mini-FLOTAC.** This figure plots the probability of correctly identifying reduced therapeutic efficacy ($prob_{reduced}$) as a function of the mean total survey costs ($cost_{total}$) across six survey designs for the three soil-transmitted helminth species and four levels of endemicity (see Table 1).
(PDF)

**S7 Fig. The probability of correctly detecting presence of reduced drug efficacy and the total survey cost for six survey designs across four levels of endemicity when deploying FECPAK$^{G2}$.** This figure plots the probability of correctly identifying reduced therapeutic efficacy ($prob_{reduced}$) as a function of the mean total survey costs ($cost_{total}$) across six survey designs for the three soil-transmitted helminth species and four levels of endemicity (see Table 1).
(PDF)

## Acknowledgments

First and foremost, the authors would like to express their gratitude towards all children, their parents, the school teachers and principals that participated in this study. Second, we wish to specifically thank all the people that provided the necessary laboratory and logistic support in each of the four different sampling sites. This work would not have been possible without their willful participation and assistance.

## Author Contributions

**Conceptualization:** Luc E. Coffeng, Marco Albonico, Sake J. de Vlas, Jennifer Keiser, Jozef Vercruysse, Bruno Levecke.

**Data curation:** Johnny Vlaminck, Piet Cools, Adama Kazienga, Bruno Levecke.

**Formal analysis:** Luc E. Coffeng, Johnny Vlaminck, Piet Cools, Adama Kazienga, Bruno Levecke.

**Funding acquisition:** Marco Albonico, Alan Fenwick, Michael French, Jennifer Keiser, Gemechu Leta, Jozef Vercruysse, Bruno Levecke.

**Investigation:** Luc E. Coffeng, Johnny Vlaminck, Piet Cools, Shaali M. Ame, Mio Ayana, Daniel Dana, Leonardo F. Matoso, Simone A. Pinto, Bruno Levecke.

**Methodology:** Luc E. Coffeng, Johnny Vlaminck, Piet Cools, Mio Ayana, Giuseppe Cringoli, Jennifer Keiser, Bruno Levecke.

**Project administration:** Bruno Levecke.

**Resources:** Marco Albonico, Shaali M. Ame, Mio Ayana, Alan Fenwick, Michael French, Stefanie Knopp, Gemechu Leta, Maria P. Maurelli, Antonio Montresor, Greg Mirams, Zeleke Mekonnen, Rodrigo Corrêa-Oliveira, Laura Rinaldi, Somphou Sayasone, Peter Steinmann, Eurion Thomas.

**Software:** Luc E. Coffeng, Matthew Denwood, Greg Mirams, Eurion Thomas.

**Supervision:** Johnny Vlaminck, Piet Cools, Sake J. de Vlas, Bruno Levecke.

**Validation:** Luc E. Coffeng, Johnny Vlaminck, Piet Cools, Matthew Denwood, Sake J. de Vlas, Bruno Levecke.

**Visualization:** Luc E. Coffeng, Johnny Vlaminck, Piet Cools, Bruno Levecke.

**Writing – original draft:** Luc E. Coffeng, Johnny Vlaminck, Piet Cools, Bruno Levecke.

**Writing – review & editing:** Luc E. Coffeng, Johnny Vlaminck, Piet Cools, Matthew Denwood, Marco Albonico, Shaali M. Ame, Mio Ayana, Daniel Dana, Giuseppe Cringoli, Sake J. de Vlas, Alan Fenwick, Michael French, Adama Kazienga, Jennifer Keiser, Stefanie Knopp, Gemechu Leta, Leonardo F. Matoso, Maria P. Maurelli, Antonio Montresor, Greg Mirams, Zeleke Mekonnen, Rodrigo Corrêa-Oliveira, Simone A. Pinto, Laura Rinaldi, Somphou Sayasone, Peter Steinmann, Eurion Thomas, Jozef Vercruysse, Bruno Levecke.

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
