## [Decision Letter · Decision Letter 0]

2 Mar 2023

Dear Dr. Coffeng,

Thank you very much for submitting your manuscript "A general framework to support cost-efficient fecal egg count method and study design choices for large-scale STH deworming programs – monitoring of therapeutic drug efficacy as a case study" for consideration at PLOS Neglected Tropical Diseases. As with all papers reviewed by the journal, your manuscript was reviewed by members of the editorial board and by several independent reviewers. The reviewers appreciated the attention to an important topic. Based on the reviews, we are likely to accept this manuscript for publication, providing that you modify the manuscript according to the review recommendations. 

Sincerely,

David Joseph Diemert, M.D.

Academic Editor

Cinzia Cantacessi

Section Editor

Reviewer's Responses to Questions

**Key Review Criteria Required for Acceptance?**

**Methods**

-Are the objectives of the study clearly articulated with a clear testable hypothesis stated?

-Is the study design appropriate to address the stated objectives?

-Is the population clearly described and appropriate for the hypothesis being tested?

-Is the sample size sufficient to ensure adequate power to address the hypothesis being tested?

-Were correct statistical analysis used to support conclusions?

-Are there concerns about ethical or regulatory requirements being met?

Reviewer #1: In the paragraph starting on line 184, the authors state that the stool samples were first grouped in batches of ten samples and then all samples were homogenized by stirring. It was unclear to me if the 10 samples from different individuals were mixed all together. When reading supplementary information 1, it became more clear to me that the samples were not mixed, but processed as individual samples in batches of 10. For clarity, it would be better if that was made clear in this paragraph of the main manuscript. Also, that subsamples from each of the individual samples were then processed using the three FEC methods.

Reviewer #2: The background and aims of the study, i.e. to provide an novel framework for cost-efficient study design to examine soil-transmitted-helminth monitoring and evaluation of deworming program efficacies. In my view the objectives were addressed appropriately and by considering key parameters including different diagnostic methods, parasite species, field-relevant different epidemiological scenarios and survey designs. The study population is clearly presented and appropriate to test the hypothesis that by comparing the above listed factors evidence-based recommendations can be developed for cost-efficient studies monitoring the efficacy of deworming programs. I don't feel competent to comment on the used statistical methods. I can't see any ethical or regulatory concerns.

Reviewer #3: Yes

Reviewer #4: Comments attached

**Results**

-Does the analysis presented match the analysis plan?

-Are the results clearly and completely presented?

-Are the figures (Tables, Images) of sufficient quality for clarity?

Reviewer #1: The paragraph in lines 403 to 419 is confusing. Maybe some of the examples can be better displayed in table form?

Line 445, the word "redrawing" sound like you drew the graphs instead of plotting the ascaris and trichuris data.

Reviewer #2: The study results are clearly and comprehensively presented, the figures and tables are of appropriate quality, correct and informative. The presented data match the analysis plan.

Reviewer #3: Yes

Reviewer #4: Comments attached

**Conclusions**

-Are the conclusions supported by the data presented?

-Are the limitations of analysis clearly described?

-Do the authors discuss how these data can be helpful to advance our understanding of the topic under study?

-Is public health relevance addressed?

Reviewer #1: No comments.

Reviewer #2: The interpretation of the data is clear and comprehensible. The limitations (and strengths) of the study are discussed in detail in a specific paragraph of the manuscript. The whole concept of the study is to optimize the planning and conduct of deworming surveys, which also demonstrates its public health relevance.

Reviewer #3: Yes

Reviewer #4: Comments attached

**Editorial and Data Presentation Modifications?**

Reviewer #1: In the title, the word method is missing an "s".

It´s not clear throughout the manuscript what is the exact range of subjects enrolled that were considered. In the abstract its states 10-2,000 (line 57), but in the method section (line 292) it says for a survey design a range of 100 - 5,000 was used.

Line 109 of the introduction, the word "those" is incorrectly used in this sentence.

Reviewer #2: Minor issues:

Figure 1 with respect to the Mini-FLOTAC flow chart in my view it should not read Accumulation but Flotation.

Concerning the calculation of the required time to analyse a stool sample with one of the three methods I wonder why for the FECPAG method time for counting the eggs is considered? As the method is being used this is not done in the lab but online by a technician from Techion Group Ltd.? This should be explained if not also considered with respect to the time calculations.

Reviewer #3: Y

Reviewer #4: Comments attached

**Summary and General Comments**

Reviewer #1: The article presented by the authors is relevant to the field of public health and provides useful information for STH deworming programs. Through a detailed analysis on the performance, time, and cost of three different FEC techniques they provide useful recommendations that can be readily applied. Therefore, I recommend publication of the article after a few minor revisions have been addressed.

Reviewer #2: This is a very important and potentially most relevant paper for all those interested in the monitoring of STH deworming programs. I find it very clearly presented and the data novel, convincing as well as significant.

Reviewer #3: (No Response)

Reviewer #4: Comments attached

PLOS authors have the option to publish the peer review history of their article (what does this mean?). If published, this will include your full peer review and any attached files.

Reviewer #1: No

Reviewer #2: No

Reviewer #3: No

Reviewer #4: No

Figure Files:

Data Requirements:

Reproducibility:

References

---

## [Editor Report · Decision Letter 1]

13 Apr 2023

Dear Dr. Coffeng,

We are pleased to inform you that your manuscript 'A general framework to support cost-efficient fecal egg count methods and study design choices for large-scale STH deworming programs – monitoring of therapeutic drug efficacy as a case study' has been provisionally accepted for publication in PLOS Neglected Tropical Diseases.

Best regards,

David Joseph Diemert, M.D.

Academic Editor

Cinzia Cantacessi

Section Editor

---

## [Editor Report · Acceptance letter]

12 May 2023

Dear Dr. Coffeng,

We are delighted to inform you that your manuscript, "A general framework to support cost-efficient fecal egg count methods and study design choices for large-scale STH deworming programs – monitoring of therapeutic drug efficacy as a case study," has been formally accepted for publication in PLOS Neglected Tropical Diseases.

Best regards,

Shaden Kamhawi

co-Editor-in-Chief

Paul Brindley

co-Editor-in-Chief
